# Experimental Investigation of Relative Localization Estimation in a Coordinated Formation Control of Low-Cost Underwater Drones

**DOI:** 10.3390/s23063028

**Published:** 2023-03-10

**Authors:** Thierry Soriano, Hoang Anh Pham, Valentin Gies

**Affiliations:** 1COSMER Laboratory, University of Toulon, 83041 Toulon, France; 2IM2NP Laboratory, University of Toulon, 83041 Toulon, France

**Keywords:** low-cost underwater drones, coordinated formation control, robot operating system, vision-based navigation, relative localization estimation, extended Kalman filter (EKF)

## Abstract

This study presents a relative localization estimation method for a group of low-cost underwater drones (l-UD), which only uses visual feedback provided by an on-board camera and IMU data. It aims to design a distributed controller for a group of robots to reach a specific shape. This controller is based on a leader–follower architecture. The main contribution is to determine the relative position between the l-UD without using digital communication and sonar positioning methods. In addition, the proposed implementation of the EKF to fuse the vision data and the IMU data improves the prediction capability in cases where the robot is out of view of the camera. This approach allows the study and testing of distributed control algorithms for low-cost underwater drones. Finally, three robot operating system (ROS) platform-based BlueROVs are used in an experiment in a near-realistic environment. The experimental validation of the approach has been obtained by investigating different scenarios.

## 1. Introduction

In recent years, the systems and control community has been actively researching distributed coordination of various vehicles, such as unmanned aerial vehicles (UAV), unmanned ground vehicles (UGV), and unmanned underwater vehicles (UUV). In order to achieve a cooperative group performance, the distributed approach has several benefits, particularly with regard to low operational costs, fewer system requirements, high robustness, strong adaptivity, and flexible scalability. In general, distributed coordination research has three main problems: (i) distributed tracking control, (ii) navigation and localization, and (iii) the ability to deploy and test on real robots.

(i) Distributed tracking control: The objective of designing a distributed controller for a group of autonomous vehicles is to help them make decisions by themselves in a coordinated group task. Here, coordination refers to the close relationship between all vehicles in the group, where information sharing plays a central role. This past decade has seen intense research on distributed control of multi-robots under different aspects, such as consensus over switching network topology [1,2], consensus with delays [1,3,4], optimal consensus [5,6] sampled-data consensus, adaptive consensus [7], quantized consensus, second-order consensus [8], the consensus of generic linear agents [9,10], and consensus with multiple leaders [11]. Furthermore, under conditions of constrained radio waves, leading to limited information sharing (especially in underwater conditions), the design of distributed control algorithms has many challenges. Interested readers are referred to survey papers [12,13] for excellent reviews of the progress made before 2017 in the multi-agent coordination problem. In our previous research [14], we also demonstrated and simulated on ROS/Gazebo a complete control architecture for a group of AUVs that can maintain formation, as well as avoid collisions and change the distance among them when crossing narrow areas.

(ii) Navigation and Localization: This is a challenging problem in the development of autonomous vehicles. Navigation is to guide the robot from one point to another. Localization is how well the drone localizes itself within a map or with another robot. In particular, the problem of underwater navigation and localization is even more challenging. In [15], a review of an AUV navigation and localization method was presented. These methods can be divided into three categories: inertial/dead reckoning, acoustic transponders and modems, and geophysical. Most recently, in [16], a robot swarm relative localization method was studied, and in [17], a review of localization, navigation, and communication of UUVs for collaborative missions is presented. In [18], the fusion of an inertial sensor and a vision is introduced. The inertial sensor has six degrees of freedom (6-DoF): 3-axis of an accelerometer and the 3-axis of a gyroscope. The goal is to determine a low-cost and accurate position for an autonomous mobile robot. In [19], the research is focused on the tracking of an AUV with three light beacons to transmit their IDs using a wide FOV camera on a leader AUV. If three light beacons are captured on the image, it allows for estimating the target AUV’s pose with respect to the camera. The distance between the AUVs was estimated by using the geometry of the relative positions of the beacons on the image and the beacon’s location on the target AUV. In [20], the relative localization of mobile robots based on multiple ultra-wideband ranging measurements was studied and applied to a group of ground mobile robots. In addition, there is a technique being researched and developed called SLAM (simultaneous localization and mapping). This technique is the process by which an autonomous robot constructs a map of its environment and simultaneously locates itself in this environment. Some of the most popular categories of SLAM approaches are EKF-SLAM [21,22], FastSLAM [23], and GraphSLAM [24]. However, these methods also require a high number of sensors with high costs as well as the high computing power of embedded systems.

(iii) Experimental and numerical investigation: Actual experiments still have many parameters that are not predefined by the simulation. In addition, there are limitations in both technology and cost-effectiveness, especially for autonomous underwater vehicles. In [25], the problem of pipeline following for AUVs was studied by using a monocular camera. In [26], a robust multi-robot convoying approach that relies on visual detection of the leading agent was presented, thus enabling target following in unstructured 3D environments. Moreover, an approach that uses vision and a convolutional neural network for an autonomous underwater vehicle to avoid collisions while observing objects was studied in [27]. In addition, the EC MORPH project [28,29] also studied formation control and tested it on real robot models which were equipped with an ultra-short baseline (USBL) and an underwater acoustic communication system.

Motivated by using low-cost underwater drones with low-cost sensors in the condition of limitation of communication and localization, this study:Proposed an effective prototype using four LEDs for the low-cost underwater drones to be able to determine the relative position between robots and an experimental evaluation of the algorithms for the low-cost underwater drones, which has not yet been investigated to the authors’ knowledge.Proposed to use an EKF to estimate the position of the underwater vehicle follower-leader in case the robot is out of view of the camera. Compared with [30], the dynamic adaptive Kalman filter was used to navigate a group of AUVs in cases with and without GNSS. We implemented EKF for underwater vehicles and used only vision and IMU data. The use of cameras is low cost compared to positioning methods using sonar technology, which according to article [15], costs thousands to hundreds of thousands of dollars.Tested coordination control algorithms (i.e., formation control) in real environment conditions.

Figure 1 introduces the test scenario in this paper. In this case, we used three low-cost robots, in which it is assumed that a leader robot can be controlled to follow a given trajectory, and two follower robots automatically move to form a formation and follow the leader robot.

## 2. Related Work

### 2.1. Low-Cost Underwater Drone Modeling

The equation of the dynamics of the low-cost underwater drone *i* can be remodeled into a state space model of the form:(1)$${\dot{x}}_{i}=A_{i}x_{i}+B_{i}\left[\tau_{hi}+f_{i}\left(x_{i}\right)+w_{i}\left(t\right)\right]$$
where $x_{i}\in R^{6}$ is the state vector of the robot *i*;

$f_{i}\left(x_{i}\right)\in R^{3}$ is the unknown uncertainty of the robot;

$w_{i}\left(t\right)\in R^{3}$ represents the corresponding unknown disturbance;

$A_{i}\in R^{6\times 6}$ and $B_{i}\in R^{6\times 3}$ are known matrices and $\left(A_{i},B_{i}\right)$ is assumed to be stabilizable.

According to [31], a local feedback controller is chosen:(2)$$\tau_{hi}=K_{r}x_{i}+u_{i}$$
so that $A+BK_{r}$ is to *Hurwitz*. $u_{i}\in R^{3}$ is the control input.

### 2.2. Formation Tracking Control

We have chosen the leader and follower architecture to implement the formation tracking control for underwater vehicles with low-cost criteria. In this case, the model of the leader robot is as follows:(3)$${\dot{x}}_{L}=A_{L}x_{L}+B_{L}r\left(x_{L},t\right)$$
where $x_{L}\in R^{6}$ represents the state of the reference robot and $r\left(x_{L},t\right)\in R^{3}$ represents a reference input. $A_{L}\in R^{6\times 6}$ and $B_{L}\in R^{6\times 3}$ are known matrices of the reference robot. The reference model can be considered as a virtual leader in the group, and it generates a desired trajectory for the whole group to follow.

To control the formation, we defined an auxiliary variable vector, which is a relative position tracking error $e_{i}\in R^{6}$ for l-ROV *i*, as
(4)$$e_{i}=\sum_{j\in N_{i}}a_{ij}\left(x_{i}-\delta_{i}-\left(x_{j}-\delta_{j}\right)\right)+a_{iL}\left(x_{i}-\delta_{i}-x_{L}\right)$$
where $\delta_{i}-\delta_{j}\forall i\neq j$, indicates the desired formation between robots (see Figure 2), $a_{ij}$ and $a_{iL}$ demonstrate connectivity between robots, and $x_{L}\in R^{6}$ represents the state of the reference robot, which is defined in Equation (Equation 3).

A distributed adaptive training tracking control was then designed based on the consensus tracking protocol and the feedback linearization technique, which is given by:(5)$$u_{iFC}=c_{i}Ke_{i}$$
where $K\in R^{3\times 6}$ is the return control gain and $c_{i}\in R$ is the updated weight. Readers can find the details about formation tracking control in our previous research [14,32].

In the following experiment, we are concerned with determining the relative position between the follower and the leader robot because each robot can only be equipped with one camera. It is therefore able to detect the objects in front of it. This also leads to a limitation in that the developed control algorithms can guarantee only the constant distance between the follower robots and the leader robot but cannot guarantee the distance between the follower robots. However, this might also be improved if the robots are equipped with four cameras or a 360-degree camera.

### 2.3. Vision-Based Pose Estimation

In this section, we assume that a 3D model of the scene is available and can be estimated online. The position must be estimated by knowing the correspondences between the 2D measurements in the images and the 3D features of the model (see Figure 3).

Let us denote $F_{c}$ as the camera frame and ${}^{c}T_{w}$ as the transformation that defines the position of $F_{w}$ with respect to $F_{c}$.
(6)$${}^{c}T_{w}=\left(\begin{array}{cc} {}^{c}R_{w} & {}^{c}t_{w}\\ 0_{3\times 1} & 1 \end{array}\right)$$
where ${}^{c}R_{w}$ and ${}^{c}t_{w}$ are the rotation matrix and the translation vector defined by the camera position in a global reference frame, respectively.

The projection in perspective, ${\bar{x}}_{v}={\left(u,v,1\right)}^{⊤}$, from a point ${}^{w}X={\left({}^{w}X,{}^{w}Y,{}^{w}Z,1\right)}^{⊤}$ is given by
(7)$${\bar{x}}_{v}=K\Pi {}^{c}T_{w}{}^{w}X$$
where ${\bar{x}}_{v}$ is the perspective projection of a point X and K is the matrix of intrinsic parameters of the camera.

These parameters can be obtained by an offline calibration step. In addition,
(8)$$\Pi =\left(\begin{array}{cccc} 1 & 0 & 0 & 0\\ 0 & 1 & 0 & 0\\ 0 & 0 & 1 & 0 \end{array}\right)$$

We consider the image coordinates expressed in the normalized metric space
(9)$$x_{v}=K^{-1}{\bar{x}}_{v}$$

If we have *N* points ${}^{w}X_{i},i=1..N$ whose coordinates are expressed in $F_{w}$ are given by ${}^{w}X_{i}={\left({}^{w}X_{i},{}^{w}Y_{i},{}^{w}Z_{i},1\right)}^{⊤}$, the projection ${\bar{x}}_{vi}={\left(x_{vi},y_{vi},1\right)}^{⊤}$ of these points in the image plane is then given by:(10)$$x_{vi}=\Pi {}^{c}T_{w}{}^{w}X_{i}$$

If we know the 2D–3D point correspondences, $x_{vi}$, and ${}^{w}X_{i}$, the pose estimation ${}^{c}T_{w}$ is the solution of Equation (Equation 10). This inverse problem is known as the N-point perspective problem or PnP problem (*Perspective-n-point*). The reader can find details in [33,34].

### 2.4. Theory of Extended KALMAN Filter

A non-linear dynamic system can be described as:(11)$$x_{n}=f_{kf}\left(x_{n-1}\right)+w_{n-1}$$
where $x_{n}$ is the robot’s system state (i.e., the 3D pose) at time *n*, $f_{kf}$ is a non-linear state transition function, and $w_{n-1}$ is the noise of the process, which is assumed to be normally distributed. In addition, the form of the measures is
(12)$$z_{n}=h\left(x_{n}\right)+v_{n}$$

The Kalman filter works in a predictive loop. Once initialized, the Kalman filter must predict the state of the system at the next time step. In addition, the Kalman filter provides the uncertainty of the prediction.
(13)$${\hat{x}}_{n+1,n}=F{\hat{x}}_{n,n}+G{\hat{u}}_{n,n}$$
(14)$$P_{n+1,n}=FP_{n,n}F^{T}+Q$$

Once the measurement is obtained, the Kalman filter updates (or corrects the prediction) the uncertainty of the current state. In addition, the Kalman filter predicts the next states, and so on.
(15)$$K_{n}=P_{n,n-1}H^{⊤}{\left(HP_{n,n-1}H^{⊤}+R_{n}\right)}^{-1}$$
(16)$${\hat{x}}_{n,n}={\hat{x}}_{n,n-1}+K_{n}\left(z_{n}-H{\hat{x}}_{n,n-1}\right)$$
(17)$$P_{n,n}=\left(I-K_{n}H\right)P_{n,n-1}{\left(I-K_{n}H\right)}^{⊤}+K_{n}R_{n}K_{n}^{⊤}$$

Details of the Kalman filter variables can be found in Table 1.

Using EKF theory and the vision-based estimation presented above, combined with our previous theoretical studies [14], we introduce the implementation of relative location estimation for real robots in the next section.

## 3. Implement Relative Positioning for l-UD Followers

In previous work, we introduced a control architecture [14]. It consists of four components, which are the formation tracking control input, $u_{iFC}$, the robust control input, $u_{iR}$, the neural network control input, $u_{iNN}$, and the collision avoidance, $u_{iCA}$ (see Figure 4). We evaluated and validated these proposals on the Gazebo simulator, assuming that the underwater robots can localize and share information with other underwater robots. However, this is challenging due to the limitation of underwater radio transmission in the real environment. Therefore, we introduce an approach that uses a camera to determine the relative position between the follower and the leader robot to experiment with the complete process in this section.

This study is equivalent to the third round of the spiral model-based research methodology (see Figure 5). The first circle studies the simple simulation model and the second circle studies the complete simulation model, which was introduced in the previous study [32]. This third circle focuses on researching and testing control algorithms for low-cost underwater UAV prototypes.

Inspired by previous works [33,35] for pose estimation in computer vision, we propose the prototype of l-UD, which already has an HD USB camera with low-light performance and low-cost sensors (IMU). We then equipped the leader l-UD with four high intensity LEDs (see Figure 6). This equipment is built on the low-cost robot platform BlueROV [36], which is open source to allow for extensive customization.

Thanks to this equipment, the camera mounted on the follower l-UD is able to determine the leader l-UD in the underwater environment under low light conditions. In addition, we also placed these LEDs at a 10 cm depth inside plastic tubes to avoid light dispersion in the underwater environment (see Figure 7).

More precisely, the camera of the following robot detects four LEDs placed on the robot leader and its position. Then, it determines the relative position of the camera with respect to the center of gravity of these four LEDs. This also means determining the relative position between the follower robot and the leader robot (see Figure 8).

**Assumption** **1.**
*The assumption is that the speed of the robot leader is predefined and is within a given range.*


**Assumption** **2.**
*The loss of position between the robot follower and the robot leader only occurs for a limited time, ▵T.*


By combining the relative position obtained by the camera with the signal measured by the sensor via the EKF filter, we obtain the relative position between the l-UD follower and the leader. The details on the use of the EKF are presented in Figure 9.

The limitation of using low-cost cameras and underwater testing conditions may result in the camera being unable to detect all four LEDs continuously. As a consequence, the EKF can be used to predict the relative position between the follower robot and the leader for a short period of time. In addition, compared to machine learning algorithms, implementing EKF also allows us to take advantage of using an embedded system with limited computing power.

## 4. Experiment Setup

### 4.1. Low-Cost Underwater Drone BlueROV with LED

To perform the experiment, we used the Lumen Subsea Light, which is a blindingly bright LED light for use on ROVs and AUVs. The light outputs over 1500 lumens at 15 Watts and has a 135 degree beam angle for wide illumination in front of an ROV. Moreover, this LED has a fully dimmable control using a PWM servo signal and simple on–off control with no signal needed. In addition, while testing, we found that underwater light scattering can reduce the detection efficiency of the camera. Therefore, we also placed these LEDs in plastic tubes (with a length of 30 cm) to help the lights concentrate on one point (see Figure 10).

We carried out the tests in the experimental pool of the University of Toulon. The dimensions are length × width × depth = 10 m × 3 m × 1.5 m (see Figure 11). The objective of these experiments was to validate the algorithms that have been developed and simulated on the Gazebo simulator, which was presented in our previous study [14]. The successful testing of these algorithms on the Gazebo simulator assumes that the relative distances between the robots can be obtained through different positioning methods. However, in the actual test conditions in the laboratory, the use of the camera positioning method was the most feasible. These experiments also demonstrated that the algorithms can operate in a near-realistic environment (i.e., a fully open marine environment). The methods limitations are the camera’s ability to detect objects and the camera detecting the wrong object when the robot is very close to the water surface due to the phenomenon of light reflection.

Figure 7 shows three l-UD BlueROVs, including one leader robot equipped with four LEDs and two follower robots equipped with cameras. The cable visible in the image is only for monitoring purposes, but there is no communication between robots. The size of the four LEDs equipped on the leader of the robot is length × width = 0.42 m × 0.24 m.

### 4.2. Determination of the Function of the Real Distance and the Distance Measured by the Camera

Firstly, we would like to determine the actual distance and the distance measured with the camera between the two robots. For this purpose, we placed two robots at different fixed distances from 1.5 m to 8.5 m, respectively (see Figure 12). The distance of 8.5 m was the maximum distance at which the camera is capable of detecting the four LEDs in our test environment. Simultaneously, we also determined the distance measured by the camera.

By linearizing the obtained values (see Figure 13), we obtained a first-order function between the actual distance and the distance obtained from the camera:(18)y=20.207x+0.2249
where *x* is the distance measured by the camera and *y* is the actual distance. The objective of function (Equation 18) is to help correct between the real and measured distances.

### 4.3. The Experiment with Three Robots

Secondly, the three robots were placed in an arbitrary position in the experimental pool in the realized experimentation. However, they must fulfil the initial condition that the robot leader must be in the visible area of the two cameras on the robot followers. The robot follower detects four LEDs placed on the robot leader and then calculates the relative position between the camera and the four LEDs. Without loss of generality, we assume that the relative position between the robot follower and the robot leader is also the relative position between the robot follower camera and the four LEDs mounted on the robot leader.

The initial relative position between the BlueROV-1 tracking robot and the BlueROV leader robot is (x,y) = (1.6 m, 0.5 m). The initial relative position between the BlueROV-2 follower robot and the BlueROV leader robot is (x,y) = (1.6 m, −0.5 m). The goal is that the follower robots 1 and 2 should move automatically to ensure that the relative position between them is equal to 1.2 m.

Then, the leader robot moves along a given trajectory. In this case, it is controlled via a joystick. However, it could also be possible to adopt a vision-based approach for the l-UD leader, for example, a pipeline tracking application [25].

Simultaneously, the follower robots 1 and 2 must move and always ensure their distances from the leader robot are equal to 1.2 m. During movement, the robot depths were controlled by an appropriate PD control and built-in pressure sensor.

The objective is to maintain the l-UD BlueROV follower at a distance of 1.2 m between it and the l-UD BlueROV leader, and while the leader robot moves, the follower robots need to follow the leader robot. Moreover, they still have to maintain a distance of 1.2 m from the leader robot. Figure 11a shows the initial position of the l-UD BlueROV. Figure 11b shows the l-UD BlueROV tracking underwater robots moving in formation. Figure 11c,d show the l-UD leader moving; simultaneously, the two follower robots follow the leader robot and maintain a distance of 1.2 m.

It is also to be noted that it can guarantee only the distance between the follower robot and the leader robot, but it can not ensure a constant distance between the follower robots. Figure 14 and Figure 15 show the distance between the follower robots 1 and 2 with the leader robot. We see that these distances all converge to 1.2 m. This may prove that the follower robots can keep a constant distance from the leader robot. The red line shows the distance between the robot follower and the robot leader after using the EKF filter to combine the vision and IMU values. The blue line shows the distance between the two robots when the EKF filter is not used.

Figure 14 and Figure 15 also indicate two peaks (blue lines) that indicate that the cameras of the follower robot failed to detect the four LEDs placed on the leader robot. The objective of using the EKF is to predict the relative position between the follower and leader robots in this case. The numerical values of the EKF matrices can be found in [32,37]. The architecture of software components developed on the ROS platform can be seen in Figure A1 of Appendix A.

Figure 16 and Figure 17 show the control signals of the robot followers 1 and 2. The robots are controlled in the *x* (a blue line /ux/data) and *y* (a red line /uy/data) axes, respectively, to change the distance between the follower and the leader robot. We can observe that the rotation control signal /up/data of the robots (the light blue line) has a large change because of the limitation of the accuracy of the sensors. However, these control signals can also converge when the distance between the robots reaches the desired value.

Figure 18 shows images of the two BlueROV follower underwater robots, which were taken from the camera mounted on the leader robot.

Figure 19 shows images of the four LEDs detected on the leader robot from the BlueROV follower 1 and 2.

## 5. Conclusions

In order to complete a general approach to the distributed control of a swarm of low-cost underwater robots, which requires position information, we proposed a relative localization approach based on the fusion of vision and IMU data. Our experiments show that with three low-cost UAVs moving in the underwater environment, we are able to achieve a formation. In addition, our approach presents a solution for localizing a team of drones without digital communication and navigation sensors. The use of the EKF filter fulfils the purpose of predicting the relative position between the follower robots and the leader robot in cases where the follower robot’s camera fails to detect the four LEDs placed on the leader robot (due to the environment or the robot being out of view of the camera).

In future work, robustness problems require further study to increase the accuracy. The relative position obtained between the robots is completely due to the combination of the camera and an IMU using an EKF filter. However, the use of the cameras to localize underwater robots still presents many challenges in terms of accuracy, object detection (in this case, four LEDs), and changes in light intensity of the underwater environment.

## Figures and Tables

**Figure 1 sensors-23-03028-f001:**
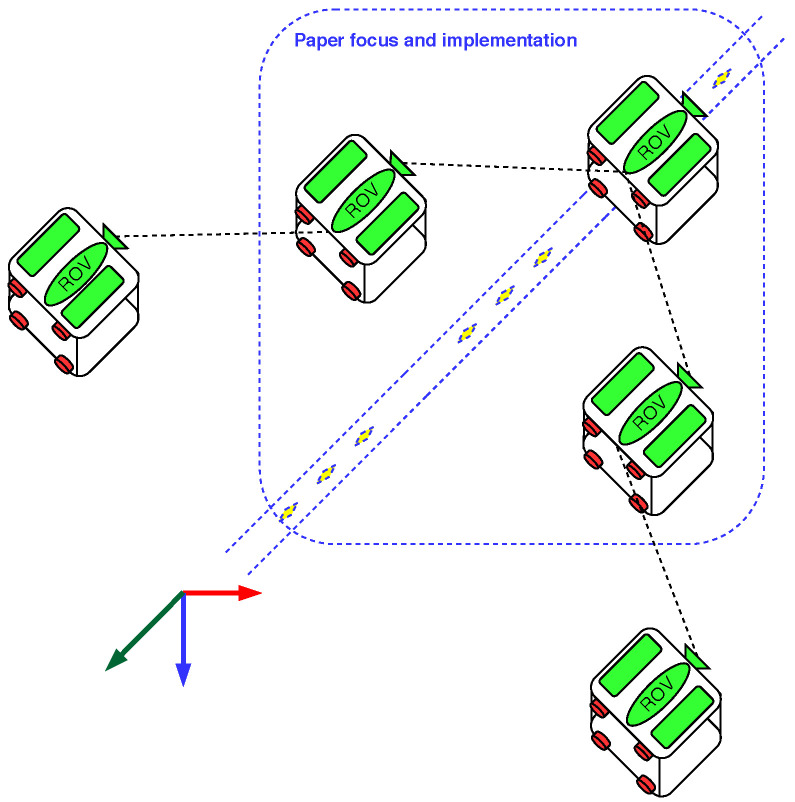
Proposal of the concept of a group of l-UDs.

**Figure 2 sensors-23-03028-f002:**
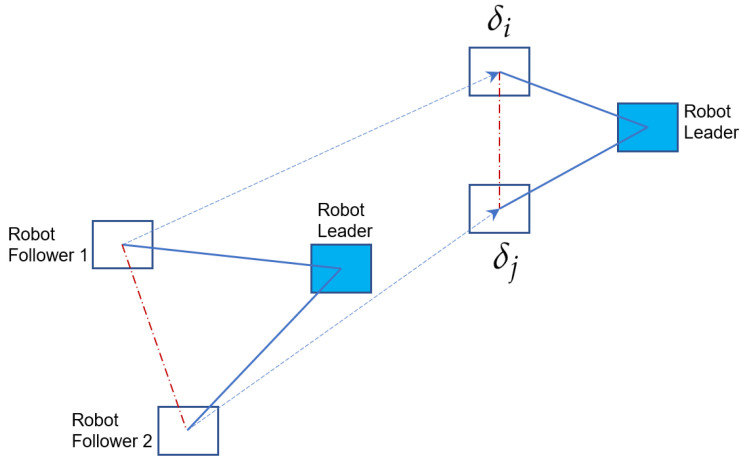
Illustration of a robot moving to make a formation. The dashed red line indicates that there is no information sharing between the two follower robots.

**Figure 3 sensors-23-03028-f003:**
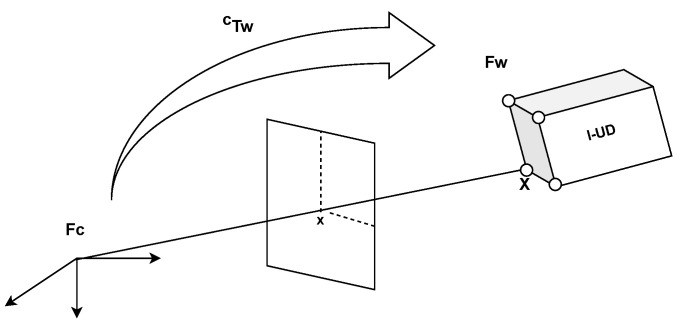
Rigid transformation ${}^{c}T_{w}$ between the world image, $F_{w}$, and the camera image, $F_{c}$, and the projection in perspective.

**Figure 4 sensors-23-03028-f004:**
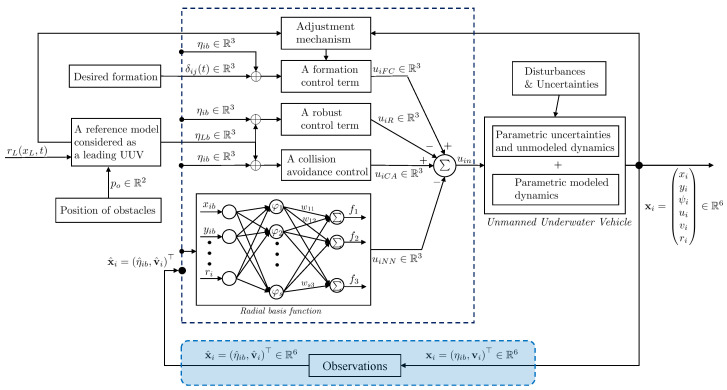
A complete diagram of the control and observation components of the low-cost underwater drone *i*.

**Figure 5 sensors-23-03028-f005:**
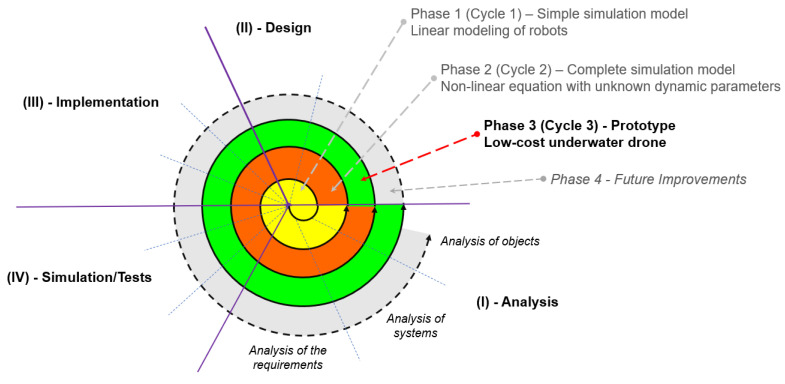
Iterative research on low-cost underwater robot systems.

**Figure 6 sensors-23-03028-f006:**
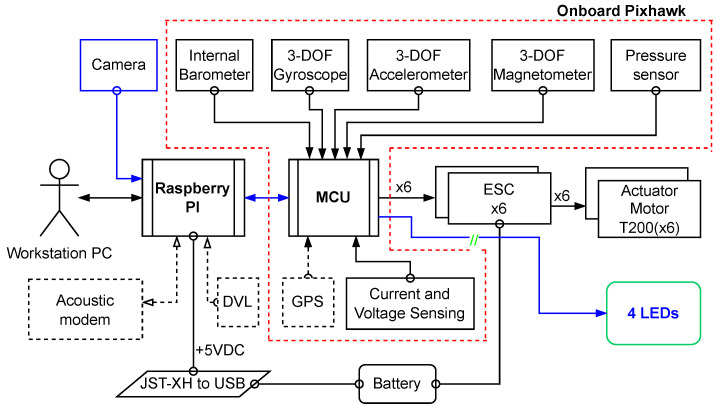
Architecture of the electronic components of the BlueROV underwater robot with the addition of four LEDs.

**Figure 7 sensors-23-03028-f007:**
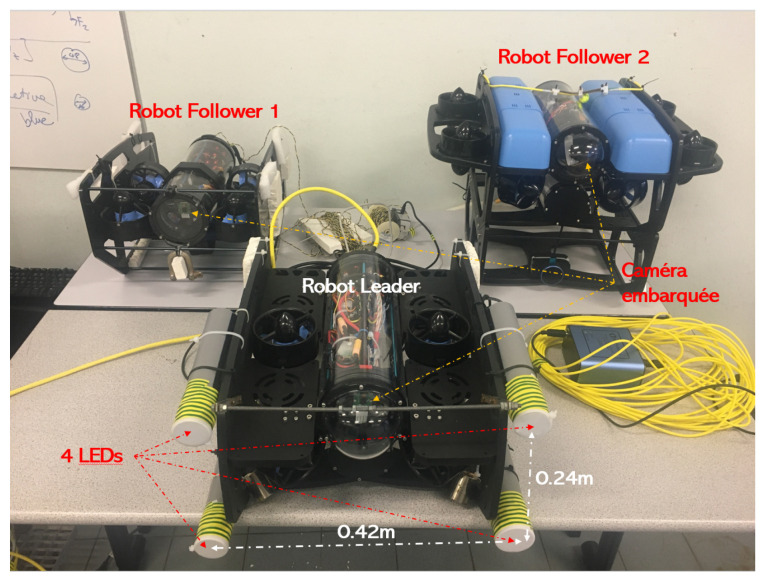
Three l-UD BlueROVs for experimentation.

**Figure 8 sensors-23-03028-f008:**
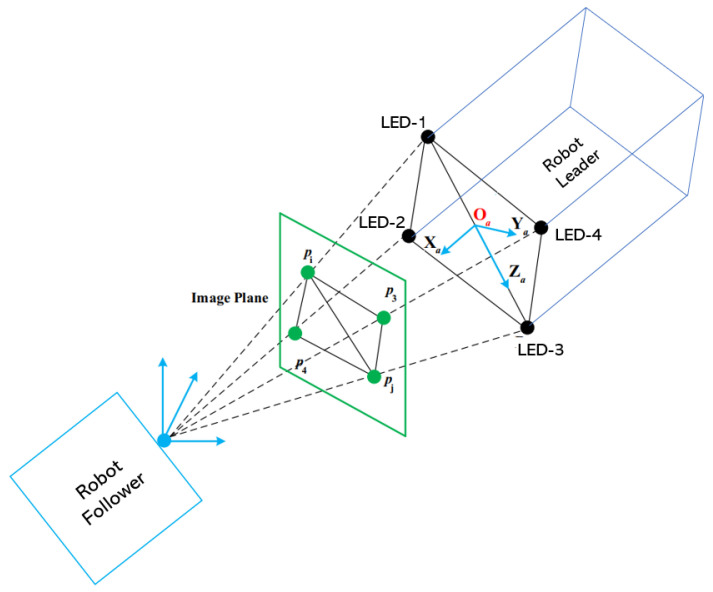
Determination of the relative position of four points using the camera.

**Figure 9 sensors-23-03028-f009:**
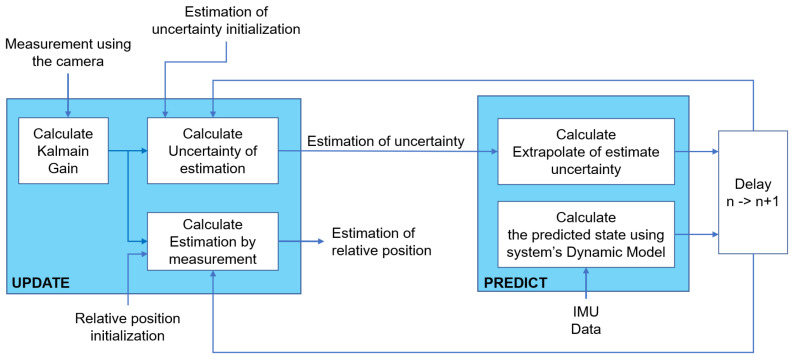
Architectural diagram of the process of fusing camera and IMU data using EKF.

**Figure 10 sensors-23-03028-f010:**
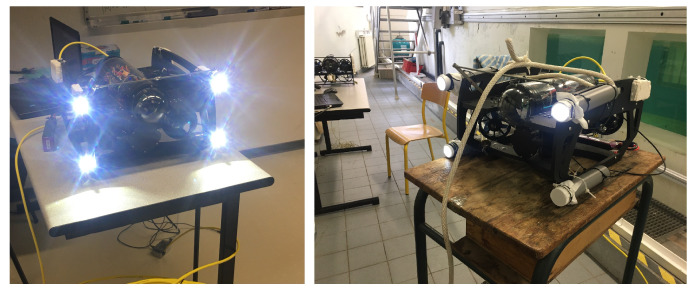
Four LEDs equipped on the follower robot.

**Figure 11 sensors-23-03028-f011:**
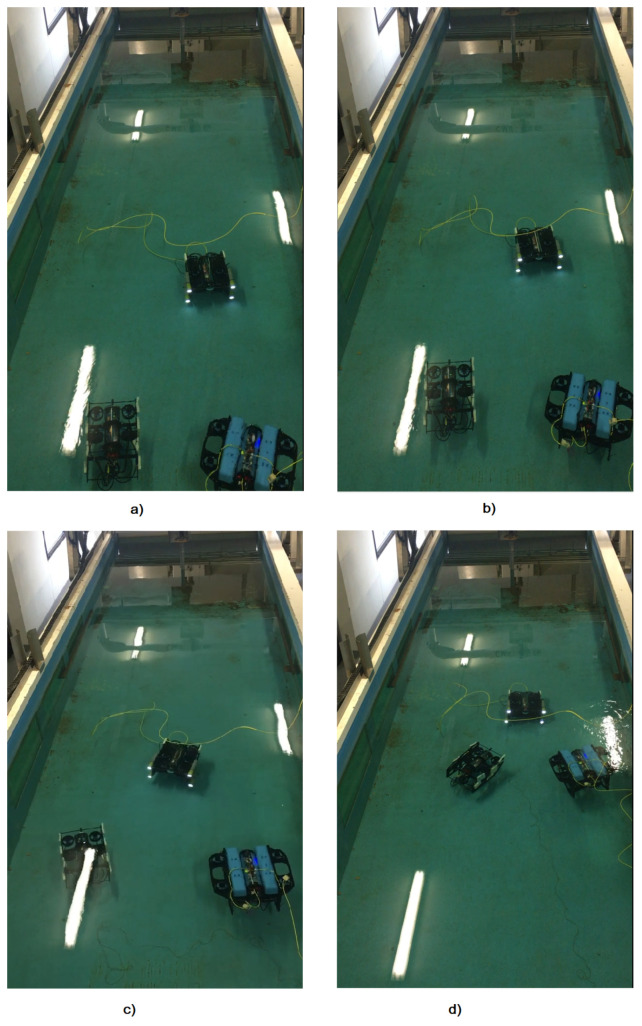
Testing of three robots forming a formation and moving according to the leading robot. (**a**) description of the initial positions of the robots. (**b**,**c**) illustration of robot group movement. (**d**) indicating the formation of the robots.

**Figure 12 sensors-23-03028-f012:**
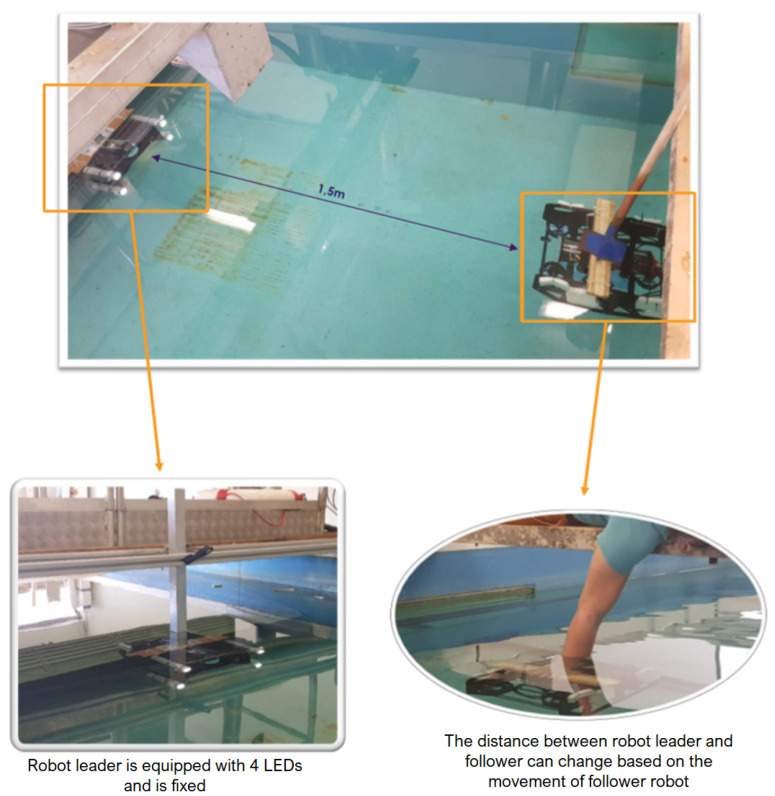
Determination of the real distance between two robots by using a camera.

**Figure 13 sensors-23-03028-f013:**
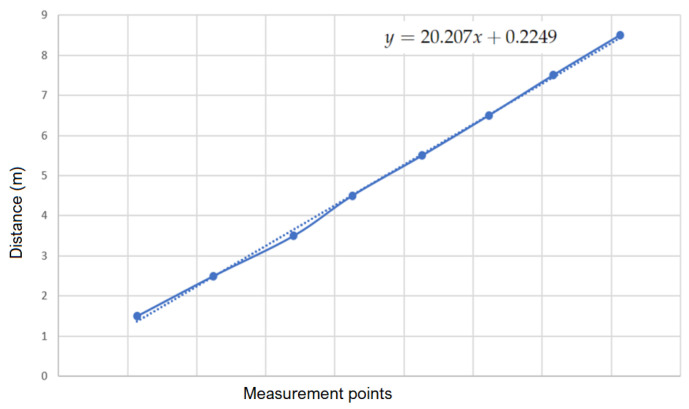
Determination of the real distance function two robots by using a camera.

**Figure 14 sensors-23-03028-f014:**
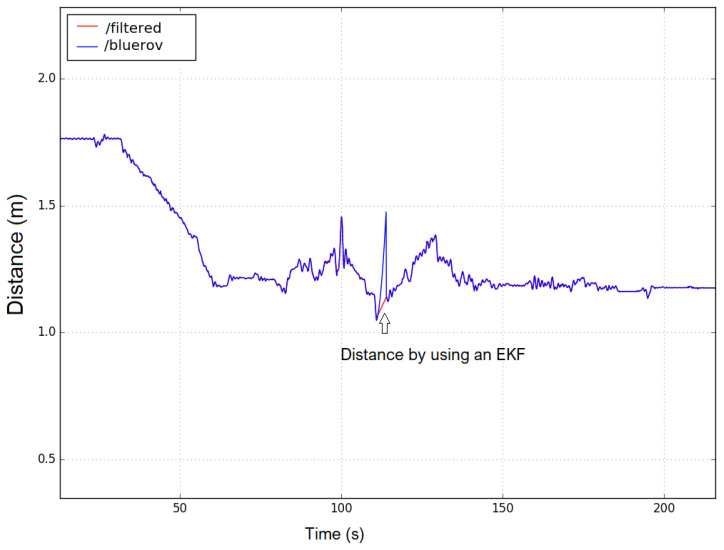
Distance between BlueROV robot follower 1 and the BlueROV robot leader.

**Figure 15 sensors-23-03028-f015:**
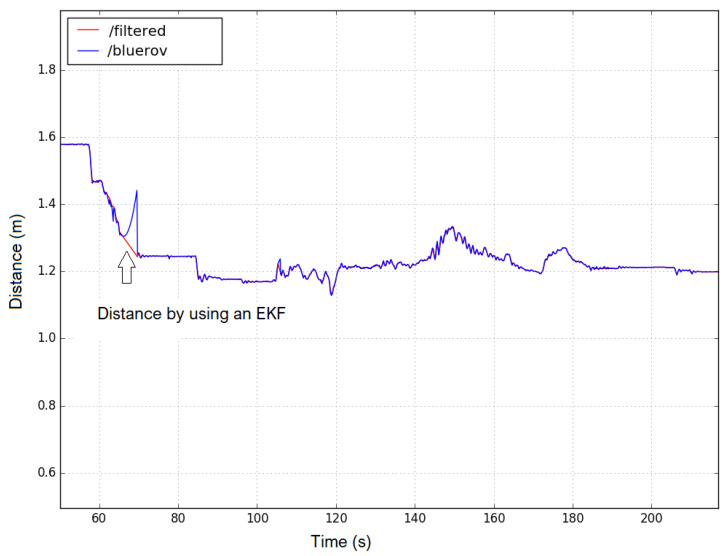
Distance between BlueROV robot follower 2 and the BlueROV robot leader.

**Figure 16 sensors-23-03028-f016:**
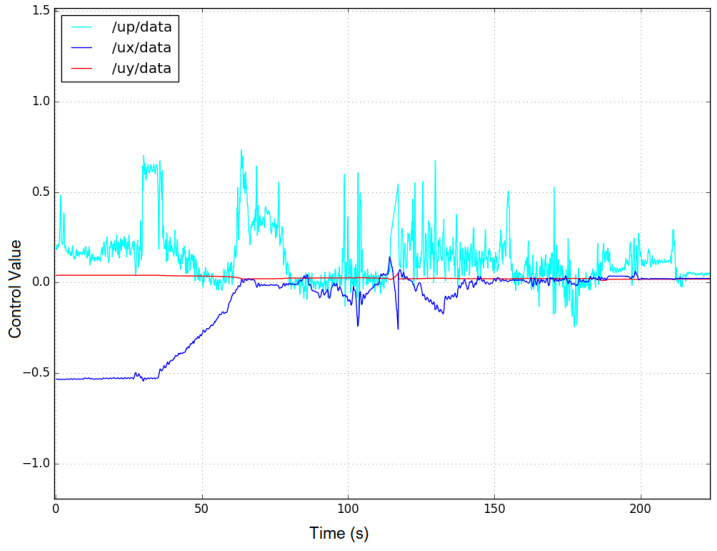
3-axis control signals for the BlueROV follower-1 robot. (The blue, red, and light blue line represent the control signal on the x-axis and y-axis and the corresponding angle control signal, respectively).

**Figure 17 sensors-23-03028-f017:**
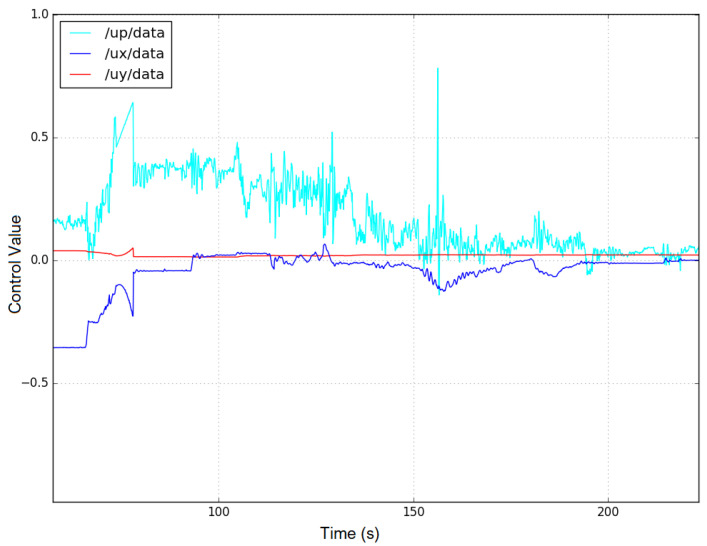
3-axis control signals for the BlueROV follower-2 robot. (The blue, red, and light blue line represent the control signal on the x-axis and y-axis and the corresponding angle control signal, respectively).

**Figure 18 sensors-23-03028-f018:**
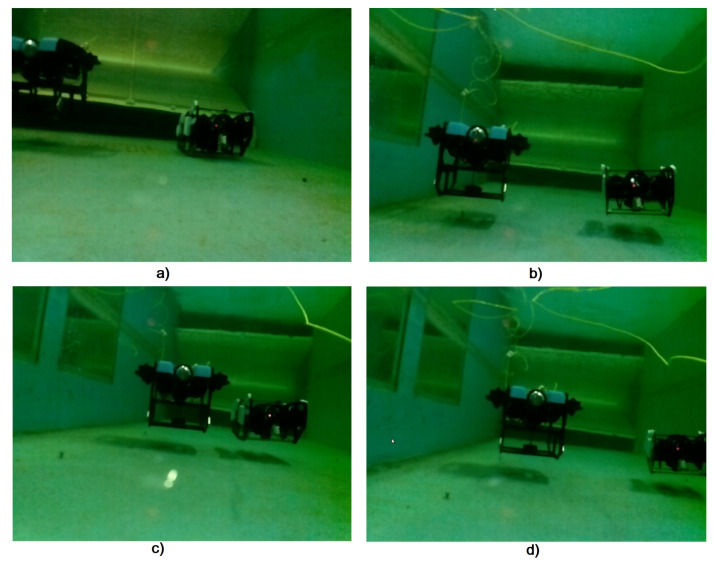
Image of the two following robots as seen from the camera of the leading robot. (**a**) description of the initial position of the robots follower. (**b**–**d**) illustration of the movement of the follower robots to make the formation.

**Figure 19 sensors-23-03028-f019:**
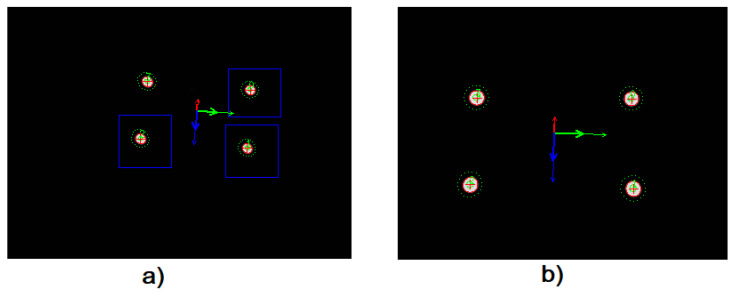
The two following robots look at the leading robot. (**a**,**b**) illustration of 4 LED detection by robot follower 1 and 2, respectively.

**Table 1 sensors-23-03028-t001:** The standard notation used for the extended Kalman filter.

Notation	Name	Dimensions
x	State vector	$n_{x}\times 1$
z	Output vector	$n_{z}\times 1$
*F*	State transition matrix	$n_{x}\times n_{x}$
u	Input variable	$n_{u}\times 1$
*G*	Matrix control	$n_{x}\times n_{u}$
*P*	Estimation of uncertainty	$n_{x}\times n_{x}$
*Q*	Uncertainty on process noise	$n_{x}\times n_{x}$
*R*	Uncertainty of measurements	$n_{z}\times n_{z}$
w	Process noise vector	$n_{x}\times 1$
v	Measurement noise vector	$n_{z}\times 1$
*H*	Observation matrix	$n_{z}\times n_{x}$
*K*	Kalman gain	$n_{x}\times n_{z}$
*n*	Discrete time index	–

## Data Availability

Not applicable.

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
