# Peer review of "Experimental Investigation of Relative Localization Estimation in a Coordinated Formation Control of Low-Cost Underwater Drones"

_sensors, 2023, doi:10.3390/s23063028_

Round 1
Reviewer 1 Report
This manuscript presents the combination of the camera and IMU using the EKF filter, to study the relative localization estimation of low-cost underwater drones based on the leader-follower control architecture. The author uses three underwater drones to test in the pool environment. The experiment proves that it can locate a group of underwater drones reach a specific shape without digital communication and navigation sensors. This manuscript provides a new method for the coordinated formation control of multiple underwater drones, but the feasibility needs to be fully demonstrated. That is, there are some major issues that need to be addressed before consideration of acceptance.
1)Under the condition of limited communication and positioning, the experiment uses low-cost underwater drones and low-cost sensors. How can the low-cost be reflected? It is understood that the use of vision for positioning should have high requirements for cameras, the cost of cameras will also increase.
2)Compared with KF, EKF increases the process to linearize the prediction and measurement parts. Please explain why EKF is fused with vision in combination with the proposed control protocol and experiment? What are the advantages (for example, Yolov5 can also achieve target tracking)?
3)Will the different heights of underwater drones affect the formation of the method proposed in this manuscript? In addition, through the components of BlueROV in Fig.3 and the experimental scenario in Fig.13, it is not clear whether the proposed method is fixed in depth? Because the difference in depth may lead to the lack of leader's LED signal by the follower.
4)The proposed experiment mentioned in Sec.4 has been simulated on the Gazebo simulator. Can the authors briefly explain the effect of the experiment in combination with the simulation results? In addition, the experiments mentioned in the manuscript are greatly affected by the underwater environment. Can the authors give a specific comparative experiment that the underwater environment is different?
5)In order to make readers clearer, please explain the x-axis, y-axis and angle expressed in Fig.11 and Fig.12 in combination with the illustration. The data in Fig.11 and Fig.12 must be different, it is suggested to discuss whether there is a correlation, because the two followers refer to the status of the same leader with the 4 LEDs.
6)Whether the initial position directly affects the formation effect, for example, if the initial position is relatively close to the formation, there may be a major problem in the robustness against noise. If the follower could not completely detect the 4 LEDs, will the formation effect be affected? To a certain extent, the experiment is deterministic to take into account more real situations.
Reviewer 2 Report
This study presents a relative position estimation method for a group of low-cost underwater drones, using only the visual feedback provided by an on-board camera and IMU data. It is based on a leader-follower architecture, without using digital communication and sonar positioning methods. It is proposed to use EKF to fuse vision data and IMU data. Three BlueROV-based Robot Operating System (ROS) platforms are used to experiment in a quasi-realistic environment in different scenarios.
His contribution may be considered low relevant and not original to the reader. The manuscript does not sound very good from a technical point of view and the readability of this manuscript is also not ok. The organisation of the manuscript is not completely focused on the topic. The overall quality is not very good.
In particular:
1) On line 25, the schematisation into three submarine robotics problems is very questionable.
2) In line 74, the equations are already known from the literature. The only new thing is the error (Eq.4 and 5) about which little is written and it is not even specified who all the terms in the equation are.
3) A rototranslation of the image is introduced (again, not original). Can the Hough transform help?
4) Section 2.4. The EKF theory is well known, better to use the space for other explanations.
5) The diagram in Fig. 3 is unnecessary, it is well known. Rather it is better to say how it calculates the distance to the image (dead reckoning is assumed to come from the IMU). Fig. 5 is also rather useless. Fig. 6, without further explanation, is only understandable if one delves deeper into the documentation and adds nothing to the document.
6) In line 142 there is something that left me very puzzled. You start from any configuration and arrive at an equilateral triangle configuration (1.2 m) of the three vehicles. But the leader-follower relationship throughout the paper (except perhaps in Eq.4) is one to one. There is no interaction, or at least it is not clear how, between followers and others. How do they not bump into each other? And what do they know about the relative configuration between them? The two followers only look at the leader and have no way of noticing each other.
7) In Figs. 9 and 10, apart from the suppression of two peaks, the graphs appear little different with and without EKF; this is unusual.
8) Fig. 11 and 12 What is in the ordinate?
The work seems to be of little value and the paper could only be accepted if reviewed in depth.
Round 2
Reviewer 1 Report
The authors have carefully revised the manuscript according the reviewer's comments. In my opinion, the revision can be accepted.
Reviewer 2 Report
Although the first answers isnot complete (My first objection referred to the questionability of the statement 'In general, distributed coordination research has three main problems: i) distributed tracking control, ii) navigation and localisation, iii) the ability to deploy and test on real robots) and furthermore there is nothing about how followers may not bump into each other, I think the work can be published.